# Widespread Detection of Fowl Adenovirus Serotype 2/11 Species D Among Cases of Inclusion Body Hepatitis–Hydropericardium Syndrome in Chickens in Egypt

**DOI:** 10.3390/microorganisms13051107

**Published:** 2025-05-12

**Authors:** Doaa M. Abdellatif, Azza A. El-Sawah, Magdy F. Elkady, Ahmed Ali, Khaled Abdelaziz, Salama A. S. Shany

**Affiliations:** 1Department of Poultry Diseases, Faculty of Veterinary Medicine, Beni-Suef University, Beni-Suef 62511, Egypt; dr.doaa.mohamed@vet.bsu.edu.eg (D.M.A.); azzasawah@yahoo.com (A.A.E.-S.); mfelkady@vet.bsu.edu.eg (M.F.E.); ahmed.ali1@vet.bsu.edu.eg (A.A.); salama.shany@vet.bsu.edu.eg (S.A.S.S.); 2Department of Animal and Veterinary Science, Clemson University, Clemson, SC 29634, USA; 3Clemson University School of Health Research (CUSHR), Clemson, SC 29634, USA

**Keywords:** fowl adenoviruses, IBH-HPS, FAdVs, phylogeny, *hexon* gene, Egypt

## Abstract

Fowl adenoviruses (FAdVs) are important emerging pathogens affecting the poultry industry in Egypt as they are the primary etiology of inclusion body hepatitis–hydropericardium syndrome (IBH-HPS) associated with severe economic losses. This study aims to identify the circulating FAdVs from cases of IBH-HPS in 5 Egyptian provinces during the period from October 2020 through September 2022. Out of the 210 examined flocks, liver samples from 66 flocks were positive for FAdVs (31.4%) using conventional polymerase chain reaction targeting loop 1 of the major *hexon* gene, with varying rates of mortality (1% to 14%). In the positive samples detected during the study, the histopathological examination revealed pathognomonic lesions of FAdVs, including basophilic and eosinophilic intra-nuclear inclusion bodies (INIBs). The percentage of FAdV positivity increased with the flock age; from samples collected at ages 1 to10, 11 to 20, 21 to 30, and >30 days of age, 10% (5/50), 25.6% (11/43), 34.3% (23/67), and 54% (27/50) were found positive for FAdVs, respectively. Notably, the positivity percentages among the flocks reared in cages were higher than for those reared in the deep litter system of housing. The gene sequencing and phylogenetic analysis of 19 strains revealed clustering into FAdV species D serotype 2/11, demonstrating that serotype 2/11 is most prevalent in the targeted Egyptian provinces during the period of the study. Several point mutations in the sequenced region among different strains were reported. These findings underscore the prevalence of FAdV and provide a basis for further research on circulating strains to develop effective control strategies.

## 1. Introduction

Fowl adenoviruses (FAdVs) are widely spread in chicken flocks worldwide. FAdVs were initially considered as opportunistic pathogens that depend on the presence of synergism with other agents, such as immune-compromising factors including infectious bursal disease (IBD) virus, chicken anemia virus (CAV), and mycotoxins, to induce the inclusion body hepatitis–hydropericardium pericardium syndrome (IBH-HPS). Recently, the existence of FAdVs as primary pathogens seems to be increasing [1,2].

The virus is a member of the family *Adenoviridae*, genus *Aviadenovirus*. Based on the molecular structure of the *hexon* protein gene and serum neutralization profiles, they are classified into five different species (FAdV-A to FAdV-E) and 12 serotypes (FAdV-A serotype 1; FAdV-B serotype 5; FAdV-C serotypes 4 and 10; FAdV-D serotypes 2, 3, 9, and 11; and FAdV-E serotype 6, 7, 8a, and 8b). FAdVs have been isolated from cases of IBH-HPS and adenoviral gizzard erosion, which are associated with severe economic losses in poultry flocks [3].

IBH-HPS outbreaks have been reported in many countries globally, such as Australia, Japan, China, Korea, Iran, India, Pakistan, Malaysia, Spain, Greece, Saudi Arabia, United Arab Emirates, Morocco, and South Africa [4,5,6,7]. All serotypes of FAdV are implicated in IBH-HPS outbreaks; however, FAdVs from cases of IBH-HPS are commonly related to FAdV-2/11 (species D) and FAdV-8a and 8b (species E) [7].

IBH-HPS is associated with severe economic losses for many reasons; firstly, as a result of the mortality rates, which generally range between 2 and 10% of the flock, although in cases of co-infection with other immunosuppressive agents such as IBDV, CAV, and REO virus, the mortality rate may reach up to 70%. The infection of broiler flocks with virulent FAdVs, such as serotype-4, has been related to significantly increased (up to 70%) mortality rates, vaccination failure due to the immunosuppressive nature of FAdVs [6,8], and the negative effects of FAdVs on feed conversion and weight gain (growth retardation).

IBH-HPS usually occurs in broiler flocks aged 3–7 weeks but may also be detected in layer flocks up to 20 weeks of age with reduced egg production and hatchability. The affected birds may die with no prominent clinical signs or exhibit non-specific signs such as the adoption of a crouching position with ruffled feathers and decreased feed intake. However, in many flocks, the infected birds have lesions relevant to IBH, with the liver and heart being the most affected organs. The liver appears pale, enlarged, and friable, with focal areas of necrosis or congested with petechial hemorrhages. Heart lesions are also seen in the form of increased volumes of pericardial fluids, which are commonly clear, straw-colored, and “jelly-like”. Anemia with icteric skin was also noticed in some birds. The lesions in the kidneys were also reported as swollen, pale, and enlarged kidneys with petechial hemorrhages [9].

In Egypt, different serotypes within different species of FAdVs were isolated from IBH-HPS outbreaks [10]. During 2019–2020, FAdV species E serotype 8a was isolated from a 35-day-old commercial broiler flock in Behira province [11]. FAdV-D 2/11 was also isolated from IBH-HPS outbreaks in Sharkia province in 2020 [12]. During a molecular investigation across the Nile Delta provinces, the dominance of species D and E serotype 8a of FAdV in broiler flocks and the emergence of new FAdV serotypes 1, 3, and 8b were reported [13]. In 2021, the new highly virulent FAdV serotype 4 species C was isolated for the first time from IBH-HPS cases encountered in a 32 days-old broiler flock in Alexandria province that suffered from a high mortality rate [14]. In Assiut province, the primary pathogenicity of FAdV serotype 2 was assessed following its isolation from broiler flocks suffered from increased mortality rates and poor performance with the absence of any immunosuppressive agents [2].

This study aims to identify the different serotypes of FAdVs associated with IBH-HPS in five Egyptian provinces (Menia, Beni-Suef, Fayoum, Giza, and Behira) during the period from 2020 to 2022.

## 2. Materials and Methods

### 2.1. Study Area

The study was conducted in five Egyptian provinces (Menia, Beni-Suef, Fayoum, Giza, and Behira) from October 2020 through September 2022 (Figure 1).

### 2.2. Sample Collection and Preparation

Liver specimens (5–10 samples per flock) were collected from 208 broiler chicken farms (aged 1 day to 6 weeks), including freshly dead and euthanized moribund chickens, one native breed, and one commercial layer flock. The vaccination program of the broiler breeders was unavailable. The liver tissue samples were transferred to the laboratory on ice and stored at −80 °C for further processing. The collected liver specimens for each farm were pooled and homogenized using a sterile porcelain mortar and pistol. A sterile saline solution was used to prepare the 20% (*w*/*v*) suspensions.

Centrifugation of the suspensions was carried out for 10 min at 2000 rpm in a cooling centrifuge, and then the supernatant was collected and stored at −80 °C until further use [11]. Specimens from the liver (5 mm thickness) were also preserved in the Bouin solution for histopathological examination. The tissues were dehydrated, cleared, embedded in paraffin, and sectioned via microtome into 4 µm-thick sections. The histopathological sections were stained with hematoxylin and eosin stain and examined under a light microscope for the detection of inclusion bodies and other pathological changes [15].

### 2.3. Molecular Detection of FAdVs

Viral nucleic acid was extracted using a genomic DNA Mini kit (Geneaid, New Taipei City, Taiwan) according to the manufacturer’s instructions. The DNA extracts were kept at −20 °C for further analyses. A polymerase chain reaction (PCR) was performed using the sense *hexon* A (5′-CAARTTCAGRCAGACGGT-3′) and antisense *hexon* B (5′-TAGTGATGMCGSGACATCAT-3′) primers for the detection of FAdVs. The primers flank a region of 897 bp in positions 144–161 for primer *hexon* A and over positions 1040–1120 for primer *hexon* B [16]. DNA amplifications were carried out in a total volume of 50 μL containing 8 μL of viral DNA, 2 μL of a 20 pmol/µL concentration from each primer, 25 μL of ABT 2X PCR mix (Applied Biotechnology, New Cairo, Egypt), and 13 μL of PCR-grade water. The reactions were run in a Multigene Gradient TC9600-G-230V thermal cycler (Labnet International, Edison, NJ, USA) as follows: an initial denaturation step at 95 °C for 5 min; 35 cycles of amplification (each amplification cycle included secondary denaturation at 95 °C for 30 sec, annealing at 60 °C for 45 sec, and extension at 72 °C for 1.5 min), and a final extension step at 72 °C for 10 min. The amplified PCR products were analyzed via electrophoresis in a 1% agarose gel, stained with ethidium bromide 0.5 μg/mL [17], and visualized via UV transillumination.

### 2.4. Sequencing and Sequence Analysis of the L1 Region of the Hexon Gene

The amplified PCR products of size 897 bp were excised and purified using a GeneJet Gel Extraction Kit (GeneJet, Thermoscientific, Vilnius, Lithuania). The purified PCR products were shipped and subjected to sequencing reactions using a Big Dye Terminator v3.1 Cycle Sequencing Kit (Applied Biosystems, Foster City, CA, USA) according to the manufacturer’s specifications at Macrogen Korea. A multiple-nucleotide sequence alignment was assembled and analyzed with a representative reference to the FAdVs strains’ *hexon* gene sequences obtained from the GenBank database (http://www.ncbi.nlm.nih.gov/) using BioEdit software version 7.0 with the ClustalW alignment algorithm, with which the percentage identity matrices between different virus sequences were determined. Neighbor-joining phylogenetic trees were constructed using the distance-based method in MEGA software version 11.

## 3. Results

### 3.1. Clinical Cases and Gross Lesions

From October 2020 to September 2022, 210 flocks (208 broilers, one native breed, and one commercial layer flock) suffering from IBH or HPS in five Egyptian provinces were examined. The distribution and mortality rates among the different flocks are summarized in Appendix A. The most prominent gross lesions observed were enlarged hemorrhagic livers with necrotic foci and the accumulation of straw-colored serous fluid within the pericardial sac. Other associated lesions within other parenchymatous and lymphoid organs were observed, such as enlarged hemorrhagic kidneys and spleens with necrotic patches (Figure 2). The distribution and positivity percentages of IBH-HPS infection with different ages and systems of housing in the investigated chicken flocks are shown in Table 1 and Table 2.

### 3.2. Histopathological Findings

The hepatic tissue appeared to be suffering from coagulative necrosis of the hepatocytes with prominent hemosiderin pigments, as well as hemorrhage; the infected cells appeared to be highly degenerated and vacuolated, and other hepatocytes appeared to be highly destructed, forming necrotic foci. Different forms and shapes of basophilic and acidophilic intranuclear inclusion bodies (INIBs) were prominent in the liver sections. The central veins and portal vessels were severely dilated and congested, with massive infiltration by lymphocytes and plasma cells around the portal area. Notably, massive fibrosis around the central vein and the portal area was evident (Figure 3).

### 3.3. Virus Detection via PCR

A PCR was performed using fowl adenovirus *hexon* gene-specific primers targeting a PCR product of 897 bp (Figure 4) on the collected liver tissues. Of 210 examined flocks, 66 were positive for FAdVs (31.4%). The percentages of flocks found positive for FAdVs from samples collected at ages 1 to 10, 11 to 20, 21 to 30, and >30 days of age were 10% (5/50), 25.6% (11/43), 34.3% (23/67), and 54% (27/50), respectively. The percentages of FAdVs-positive flocks from samples collected from deep litter and cage system farms were 26.6% (46/173) and 54.1% (20/73), respectively.

### 3.4. Gene Sequencing and Phylogenetic Analysis

The L1 regions of the *hexon* gene of the 19 selected FAdVs were sequenced, and the nucleotide sequence of each product was submitted to GenBank NCBI under various accession numbers (from PP993159 to PP993177). The 19 sequences obtained in the current study were aligned with other reference sequences of the different FAdV species (from A to E). The phylogenetic analyses and the pairwise identity matrix were created based on the alignment results. The phylogenetic analysis showed that the 19 sequenced FAdVs were clustered into fowl adenovirus species D serotype 2/11. A compressed phylogenetic tree was created based on our sequences, as well as the reference strain’s *hexon* gene and complete genome sequences from different countries that were available on the GenBank to confirm the clustering of the various FAdV species. Another phylogenetic tree expressing species D was constructed using our isolates and all of the Egyptian serotype 2/11 isolates until now (Figure 5). The genetic identity of the detected FAdV species D had a high homology (97.9%) to serotype 2 isolate accession number KT862805 from Egypt [18] and 94.9% to a serotype 2/11 isolate accession number MT975968 isolated from Poland [19] (Table 3).

### 3.5. Mutation Analysis of Amino Acid Residues

The amino acid sequences were aligned and the analysis showed several silent and non-silent mutations in the four HVRs (HVR1: n49–n243, aa17–aa81; HVR2: n244–n291, aa 82–aa97; HVR3: n337–n429, aa113–aa143; HVR4: n484–n501, aa162–aa167) [19] of the L1 region of the Egyptian and reference strains of 2/11 available on the GenBank NCBI and the sequences generated in this study (Table 4 and Appendix A).

## 4. Discussion

IBH-HPS is considered an emerging vertically transmitted disease that impacts the broiler sector and leads to significant economic losses across the globe. While all serotypes of FAdV have been associated with significant IBH-HPS outbreaks, recent reports have documented that severe IBH-HPS outbreaks are primarily associated with serotypes D (FAdV-2 and FAdV-11) and E (FAdV-8a and 8b) [6].

Histopathology, a commonly used method for diagnosing numerous IBH-HPS outbreaks in various countries, including Canada and Iraq, remains an important diagnostic tool [20,21]. Other studies have also identified PCR as a reliable diagnostic tool [22]. In the current study, we utilized both histopathology and PCR targeting a 897 bp length of the *hexon* protein gene L1 region for the detection of FAdV in samples collected from chicken flocks [16]. The primers were used for partial *hexon* gene sequencing, as they flank a very important frame of the gene, which contains full-length 3HVR (2nd, 3rd, and 4th HVR) and a part of the 1st HVR of the *hexon* protein gene [19]. It has previously been reported that the virus persisted in the liver tissue for up to 35 days post-infection (dpi), with viral DNA detectable as early as three dpi, whereas the viral nucleic acid detection in cloacal swabs was intermittent [11]. In another study, viral DNA in cloacal swabs was detected at 3 and 6 dpi only, whereas in liver tissue, it was detectable at 3 dpi and persisted for up to 13 dpi [23,24].

It is well-known that internal organs, particularly the liver, are the primary sites for FAdV replication, with the viral load and viral DNA levels reported to be higher within liver tissue than in other organs, such as the kidney and spleen [25]. Therefore, in the current study, liver tissue specimens were targeted for FAdV DNA detection rather than cloacal swabs and internal organs other than the liver. An FAdV infection was considered positive when pathological alterations were noted in the hepatic tissues from FAdV PCR positive samples. Among the pathological changes are the presence of basophilic and eosinophilic INIB in the liver tissue, coagulative necrosis, a prominent deposition of hemosiderin pigments, and a large amount of fibrosis around the central vein and portal area [26,27]. The detection of viral-associated histologic lesions and viral nucleic acids among the tested flocks at different ages (from one day old to seven weeks old) throughout the study suggests a broad age susceptibility to FAdV infection [1,27,28].

Although FAdVs are associated with high mortality rates that may reach 40% in the case of IBH and up to 80% in the case of HPS, positive flocks exhibit non-specific clinical signs [29]. In this study, varying rates (1% to 14%) of mortality were noticed in FAdV-positive flocks. In line with these observations, significant variations in mortality rates (from 1% to 30%) were previously reported in different countries such as Japan, China, and Canada [30,31], indicating that several factors might aggravate the disease [1]. Grossly, the examined flocks had variable liver lesions, including hepatic enlargement with pale pinpointed foci, being congested with round edges, liver ecchymotic hemorrhage, and in some cases the liver appeared pale with hemorrhagic patches. Effusion of the pericardium with different amounts of straw-yellow-colored fluid from 1 mL up to 15 mL may explain the noticeable decrease in the size of the heart. Additionally, hemopericardium was seen in some cases. In addition, nephrosis and an enlarged spleen with necrotic patches were noticed, as in Figure 2. These lesions represent the typical gross features of IBH-HPS, as described in previous studies [9,32]. In this study, we noticed higher positivity rates of FAdV infection among flocks reared in cages than those reared in deep litter systems, and this could be attributed to the stressors (such as the higher stocking density and rate of metabolism encountered in cages) that play a role in exacerbating the multiplication and spread of FAdVs [33,34].

Notably, three of the investigated flocks (e.g., flocks #18, #19, and #20 in Appendix A) were positive for FAdVs at day one of age, highlighting the potential of FAdVs’ early transmission, either vertically from infected broiler breeders through eggs or horizontally post-hatching [26]. Moreover, two other flocks (#57 and #62) were negative for FAdVs at day one, and positive results were confirmed for the same flocks at an older age (at 21 and 33 days). This may be attributed to the low viral load in the samples collected earlier compared to those collected later for flock #57 or due to a horizontal infection at the farm during the rearing period for flock #62 (Appendix A). Additionally, an ascending pattern of FAdV-positive flocks was observed, beginning with 10% for young flocks (1–10 days old) and reaching 54% for flocks aged over 30 days [27]. Despite some studies indicated severe degenerative changes in FAdV-infected chicks during the first week of life [35,36], the waning of the maternally derived antibodies (MDA) was reported to play an important role in the inactivation of virus replication in infected birds. If present, the MDA can delay FAdV replication and subsequent invasion of the virus into the internal organs, although it has no role in protecting against initial infection [37]. Further studies are needed to determine the role of vertically transmitted FAdVs in initiating severe outbreaks of disease.

Previous studies conducted in Egypt reported FAdV serotypes from various species, including serotype 8a [11,38] and 8b [13] species E and a new highly virulent FAdV serotype 4 species C [14], with two other different serotypes 1 and 3 belonging to different species A and B, respectively [13]. The current study only identified serotype 2/11, and it is worth mentioning that even though at least 110 sequences related to serotype 2/11 have been reported in Egypt, only one sequence of serotype 4, one sequence of serotype 1, two sequences of serotype 3, nine sequences of serotype 8a, and two sequences of serotype 8b from Egypt are available on the NCBI GenBank. These studies denote the widespread FAdV serotype 2/11 infection compared to other serotypes in Egypt [39].

Although the lengths of amino acid residues of the *hexon* protein are similar among FAdV-2/11 serotypes (950aa for each), they vary throughout other FAdV serotypes [40]. The primers used in the current study flank the most hypervariable region in the *hexon* gene, the L1 region, which is used to differentiate between the species of FAdVs, with seven hypervariable regions (HVRs 1–7) being recognized. Four HVRs are located in loop L1, two HVRs in loop L2, and the last one in loop L4; therefore, the L1 region is considered a type- or species-specific area, as it contains four HVRs, with characteristic nucleotide and amino acid sequence lengths for each FAdV species. For species 2/11, the lengths of the four HVR were as follows: HVR1: n49–n243, aa17–aa81; HVR2: n244–n291, aa 82–aa97; HVR3: n337–n429, aa113–aa143; HVR4: n484–n501, aa162–aa167 [19].

The major capsid proteins of FAdVs are efficient in triggering both cellular and humoral immune responses. The *hexon* protein, as it is the major structural component of the virus capsid and carries specific antigenic determinants on its surface, elicits the highest level of cellular or antibody response [41]. The L1 region represents one of the three variable loops (L1, L2, and L4) exposed on the outer surfaces between serotypes to form type-specific epitopes. It contains immunogenic HVRs on the surface of the *hexon* gene. HVRs from loops L1 and L2 code for type-specific antigenic determinants present on the *hexon* surface and are solely in charge of triggering type-specific neutralizing antibodies (NAbs). Furthermore, the L1 region’s HVRs load antigenic determinants unique to each serotype, especially the HVR1 [19]. The *hexon* molecule sizes differ according to the species and the serotype of FAdVs. A partial sequence analysis of the *hexon* gene of 19 samples in the current study indicated that all of them belong to FAdV-D serotype 2/11, which was also reported in many other countries, such as Canada [31], South Africa [35], China [32], Poland [19] Saudi Arabia [18], and Egypt [2,13,21,28,39].

The strains sequenced in our study shared from 99.5% upto 100% and from 98.0% upto 100% nucleotide and amino acids identity similarity between them, respectively. With the two reference strains, our sequences shared 94.2% to 94.9% and 94.7% to 95.9% nucleotide and amino acid identity similarity, respectively, with strain MT975968; and 96.7% to 97.9% and 96.2% to 97.4% nucleotide and amino acid identity similarity, respectively, with strain KT862805 (one of the only three isolates available with a full *hexon* gene sequence of Egyptian strain of FAdV 2/11) (Table 4). Differentiating serotype 2 from serotype 11 based solely on *hexon* gene sequencing is challenging, as FAdV-2 and FAdV-11 share a high degree of genetic identity and cross-neutralizing potential, and further studies are needed to address this issue. The percent identity similarity of FAdV species D detected in the present study was as high as 97.9% to serotype 2 isolate accession number KT862805 from Egypt [18] and 94.9% to a serotype 2/11 isolate accession number MT975968 isolated from Poland [19] (Table 3). Several studies were conducted to broaden the range of protection against FAdV infections in broiler flocks by the transferred MDA, a key factor in protecting the progeny until they develop age resistance to FAdV infections. These studies involved vaccines of broiler breeders containing either FAdV-11 mixed with FAdV-8a or -8b, which resulted in the development of antibodies against the related serotypes, as well as FAdV-2 and FAdV-7, as confirmed by a virus neutralization test (VNT) and challenge of the progeny [42]. In contrast, a single vaccination with live FAdV-8b raised antibodies against FAdV-8a and -8b but not against FAdV-2 or -11 [42,43]. It is, therefore, critical that FAdV be monitored continuously and that vaccine strains be evaluated and updated regularly to ensure sustained efficacy in disease prevention.

The deduced nucleotide and amino acid sequence analysis for the isolates detected in the current study with two reference strains (KT862805 and MT975968) retrieved from the GenBank NCBI revealed several silent and non-silent mutations in the four HVRs of LI of the *hexon* gene. Mutations in the sequenced region of the *hexon* protein gene were reported in all Egyptian strains available in the NCBI database and are tabulated in Table 4. The significance of the mutations reported in this study in the pathogenicity or immunogenicity of different FAdVs requires further studies.

## 5. Conclusions

Based on a sequencing analysis of loop 1 of the *hexon* gene, the most prevalent FAdV circulating in the targeted five Egyptian provinces was serotype 2/11 species D, which was associated with several outbreaks during the period 2020–2022. Further studies on the epidemiological situation, the role of vertical transmission in the spread, and the development of robust diagnostic tools and vaccines are essential for the establishment of better preventative measures for circulating FAdVs.

## Figures and Tables

**Figure 1 microorganisms-13-01107-f001:**
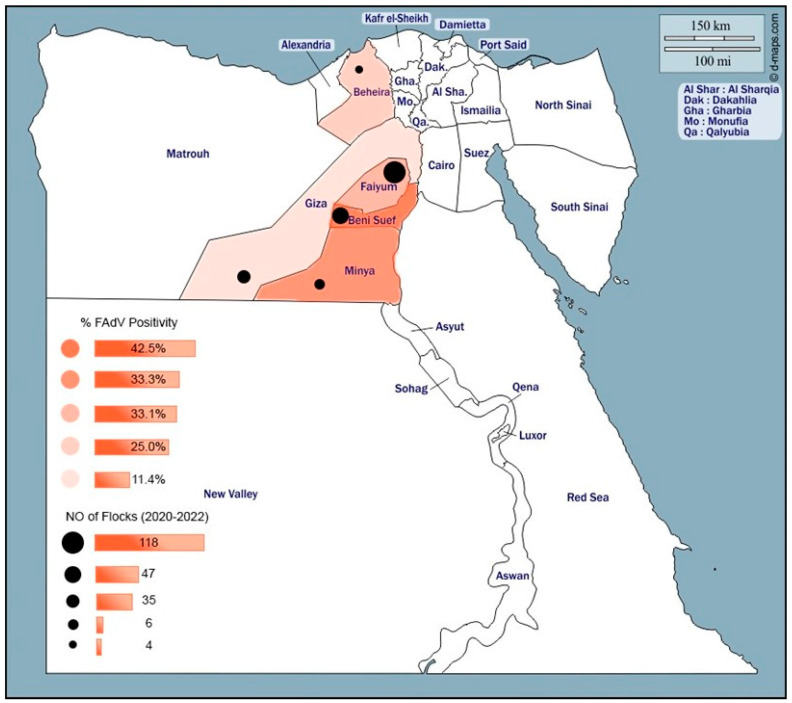
Distribution of fowl adenovirus serotype 2/11 in 5 Egyptian provinces, as well as the number of flocks collected from each province.

**Figure 2 microorganisms-13-01107-f002:**
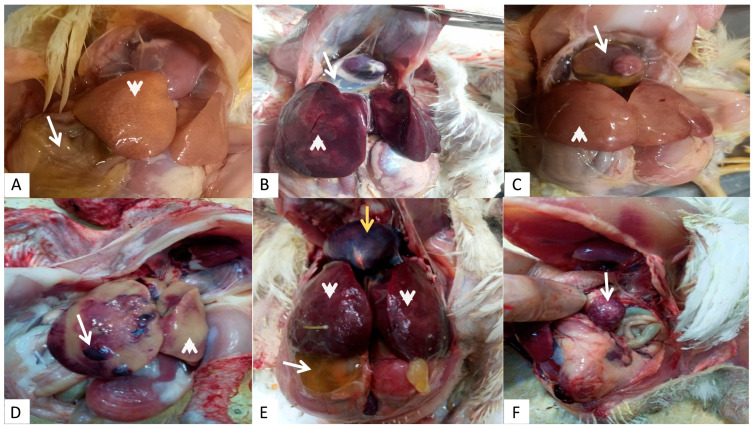
Gross pathology of chickens naturally infected with FAdVs serotype 2/11 species D (HPS): (**A**) pale liver with severe necrosis (arrowhead) and beginning of ascites (arrow); (**B**) hydropericardium (arrow) with necrosis and enlargement of the liver (arrowhead); (**C**) severe hydropericardium, heart atrophy (arrow), and pale necrotic liver (arrowhead); (**D**) enlarged liver with areas of necrosis (arrowhead) and other areas of hemorrhages (arrow); (**E**) hemopericardium (yellow arrow), enlarged congested liver (arrowhead), and ascites (arrow); (**F**) mild to moderate enlargement and areas of necrosis in the spleen (arrow).

**Figure 3 microorganisms-13-01107-f003:**
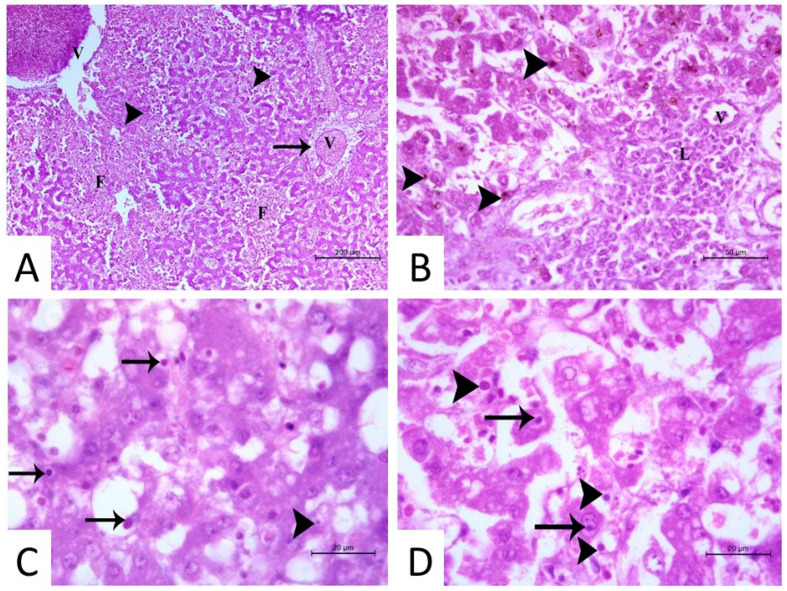
Histopathology of livers from infected broiler chickens with fowl adenovirus species D serotype 2/11. (**A**) Congestion and dilation of central veins and portal vessels (V), degeneration of hepatocytes (arrowhead) forming necrotic foci (F), and massive fibrosis around the central vein and the portal area (arrow) (H & E staining, ×100) were observed. (**B**) The hepatic tissue appeared to be suffering from coagulative necrosis of hepatocytes (arrowhead) with prominent hemosiderin pigments. A massive infiltration by lymphocytes was detected around the portal area (L) around the portal vessels (V) (H & E staining, ×400). (**C**,**D**) Different forms (basophilic and acidophilic) and shapes of intra-nuclear inclusion bodies (arrow), lymphocytes, and plasma cells (arrowhead) were detected in the infected cells, which appeared to be highly degenerated and vacuolated (H & E staining, ×1000).

**Figure 4 microorganisms-13-01107-f004:**
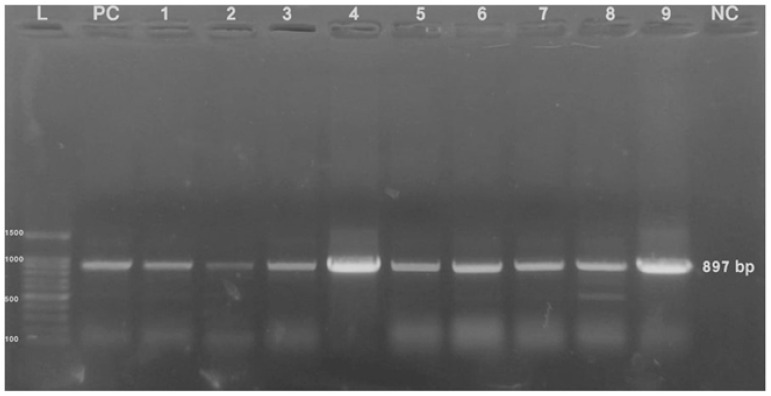
Visualization of 897 bp PCR product of the *hexon* gene of fowl adenovirus via agarose gel electrophoresis (1%) after staining with ethidium bromide. Lane L: 100bp DNA ladder; lanes 1 to 9: positive samples; lane PC: positive control; lane NC: negative control.

**Figure 5 microorganisms-13-01107-f005:**
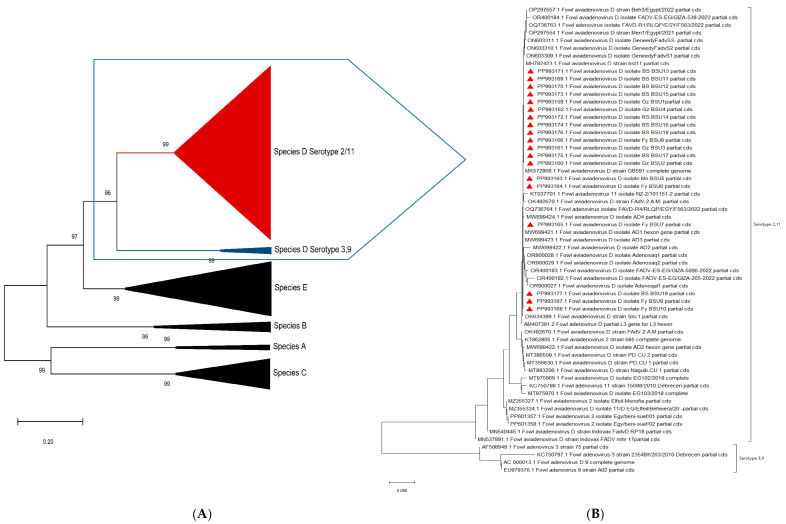
Compressed phylogenetic tree representing *hexon* gene all fowl adenovirus species (from A to E) (**A**) and detailed phylogenetic tree representing our sequenced isolates and other reference strains related to species D (**B**).

**Table 1 microorganisms-13-01107-t001:** Distribution and positivity percentages of IBH-HPS infection at different ages in investigated chicken flocks.

Province	Positivity at Different Ages (Positive/Total Tested (%))				Totals
	1–10 Days	11–20 Days	21–30 Days	>30 Days	
Menia	0/0	1/1	1/3	0/2	2/6 (33.3%)
Beni-Suef	4/17	3/4	8/16	5/10	20/47 (42.5%)
^ Fayoum **	1/22	7/28	13/36	18/32	39/118 (33.1%)
Giza ***	0/10	0/9	1/11	3/5	4/35 (11.4%)
Behera	0/1	0/1	0/1	1/1	1/4 (25%)
Subtotal	5/50 (10%)	11/43 (25.6%)	23/67 (34.3%)	27/50 (54%)	66/210 (31.4%)

^ Flocks included 117 broilers and one commercial layer. ** For samples collected from Fayoum and Giza provinces, one flock from each was negative early (at 1 and 4 days old, respectively) and positive later (at 21 and 33 days old, respectively). *** Flocks included 34 broilers and one native-breed chicken.

**Table 2 microorganisms-13-01107-t002:** Distribution and positivity percentages of IBH-HPS infection in different housing systems.

Province	System of Housing (Positive /Total Tested (%))		Totals
	Deep Litter	Cages	
Menia	2/6	0/0	2/6
Beni-Suef	13/38	7/9	20/47 (42.5%)
Fayoum	26/90	13/28	39/118 (33.1%)
Giza	4/35	0	4/35 (11.4%)
Behera	¼	0	1/4 (25%)
Subtotal	46/173 (26.6%)	20/37 (54.1%)	66/210 (31.4%)

**Table 3 microorganisms-13-01107-t003:** Nucleotide and amino acids identities of identified FAdVs in the current study.

Strain	1	2	3	4	5	6	7	8	9	10	11	12	13	14	15	16	17	18	19	20	21
Nucleotide identity %																					
1. KT862805-FAdV 2 strain 685 UK		96.7	97.6	97.8	97.8	97.4	97.7	97.8	97.4	97.7	97.9	97.9	97.7	97.6	97.4	97.6	97.7	97.6	97.8	97.6	97.8
2. MT975968 FAdV D isolate EG101/2018	96.3		94.3	94.8	94.8	94.2	94.7	94.7	94.2	94.6	94.9	94.9	94.4	94.5	94.2	94.7	94.6	94.5	94.8	94.5	94.9
3. PP993159 FAdV D isolate Gz BSU1	96.5	95		100	100	99.9	99.9	99.9	99.7	100	99.9	99.9	100	100	99.9	100	100	100	100	100	99.9
4. PP993160 FAdV D isolate Gz BSU2	96.9	95.5	99.6		100	99.9	99.9	99.9	99.6	100	99.9	99.9	100	100	99.9	99.9	100	100	100	100	99.9
5. PP993161 FAdV D isolate Gz BSU3	96.9	95.5	99.6	99.7		99.9	99.9	99.9	99.6	100	99.9	99.9	100	100	99.9	99.9	100	100	100	100	99.9
6. PP993162 FAdV D isolate Gz BSU4	96.6	95	99.6	99.6	99.6		99.7	99.7	99.6	99.9	99.7	99.7	99.8	99.9	99.7	99.9	99.9	99.9	99.9	99.9	99.7
7. PP993163 FAdV D isolate Gz BSU5	97.3	95.8	99.6	99.6	99.6	99.6		100	99.7	99.9	100	100	99.9	99.9	99.9	99.7	99.9	99.9	99.9	99.9	100
8. PP993164 FAdV D isolate Gz BSU6	97.4	95.9	99.6	99.6	99.6	99.6	100		99.7	99.9	100	100	99.9	99.9	99.7	99.9	99.9	99.9	99.9	99.9	100
9. PP993165 FAdV D isolate Gz BSU7	96.5	94.9	99.2	98.8	98.8	99.2	99.2	99.2		99.6	99.7	99.7	99.6	99.6	99.5	99.6	99.6	99.6	99.6	99.6	99.7
10. PP993166 FAdV D isolate Gz BSU8	96.7	95.2	99.6	99.6	99.6	99.6	99.6	99.6	98.4		99.9	99.9	100	100	99.9	99.9	100	100	100	100	99.9
11. PP993167 FAdV D isolate Gz BSU9	97.2	95.7	99.2	99.3	99.3	99.2	100	100	99.2	99.3		100	99.9	99.9	99.7	99.9	99.9	99.9	99.9	99.9	100
12. PP993168 FAdV D isolate Gz BSU10	97.2	95.8	99.2	99.3	99.3	99.2	100	100	99.2	99.3	99.7		99.9	99.9	99.7	99.9	99.9	99.9	99.9	99.9	100
13. PP993169 FAdV D isolate Gz BSU11	96.3	94.7	99.6	99.6	99.6	99.6	99.2	99.2	98.3	99.6	99.2	99.2		100	100	100	100	100	100	100	99.9
14. PP993170 FAdV D isolate Gz BSU12	96.5	94.9	99.6	99.6	99.6	99.6	99.2	99.2	98.3	99.6	99.2	99.2	99.6		100	100	100	100	100	100	99.9
15. PP993171 FAdV D isolate Gz BSU13	96.2	94.7	99.2	99.3	99.3	99.2	99.6	98.9	98	99.3	98.9	98.9	99.6	99.6		99.9	99.9	99.9	99.9	99.9	99.7
16. PP993172 FAdV D isolate Gz BSU14	96.5	95.1	99.6	99.3	99.3	99.6	99.2	99.6	98.8	99.3	99.3	99.3	99.6	99.6	99.3		100	100	99.9	100	99.9
17. PP993173 FAdV D isolate Gz BSU15	97.1	95.6	100	100	100	100	99.6	99.6	98.8	99.6	99.6	99.6	99.6	99.6	99.3	100		100	100	100	99.9
18. PP993174 FAdV D isolate Gz BSU16	97	95.5	100	100	100	100	99.6	99.6	98.8	100	99.6	99.6	99.6	99.6	99.3	100	100		100	100	99.9
19. PP993175 FAdV D isolate Gz BSU17	96.8	95.4	99.6	99.7	99.7	99.6	99.6	99.6	98.8	99.6	99.3	99.3	99.6	99.6	99.3	99.3	100	100		100	99.9
20. PP993176 FAdV D isolate Gz BSU18	96.7	95.2	99.6	99.6	99.6	99.6	99.6	99.6	98.5	99.6	99.3	99.3	99.6	99.6	99.3	99.6	99.6	100	99.6		99.9
21. PP993177 FAdV D isolate Gz BSU19	97.1	95.7	99.2	99.3	99.3	99.2	100	100	99.2	99.3	99.7	99.7	99.2	99.2	98.9	99.3	99.6	99.6	99.3	99.3	
Amino acid identity %																					

**Table 4 microorganisms-13-01107-t004:** An amino acid mutation analysis of all Egyptian strains related to FAdV serotype 2/11 species D.

Region	Amino Acids Position	Majority	Strains in this Study (No. 19)	Other Egyptian Strains (No. 91)
HVR1 (17–81)	58	T	--	I (1/33), -- (N = 58)
	59	R	--	H (1/34), -- (N = 57)
	60	N	--	Y (1/34), -- (N = 57)
	61	V	V (3/19), -- (N = 16)	I (1/34), L (1/34), -- (N = 57)
	62	T	T (5/19), S (1/19), -- (N = 13)	A (1 /34), -- (N = 57)
	63	T	T (11/19), -- (N = 8)	A (1/34), -- (N = 57)
	75	P	P (16/19), -- (N = 3)	A (3/34), -- (N = 57)
	78	T	T (17/19), -- (N = 2)	-- (N = 57), H (1/34), N (1/34)
	79	D	D (17/19), -- (N = 2)	G (1 /34), -- (N = 57)
HVR2 (82–97)	84	S	S (18/19), G (1/19)	S (33), -- (N = 58)
	92	N	N (19/19)	D (1/34), H (1/34), -- (N = 57)
HVR3 (113–143)	113	D	D (19/19)	G (1/75), -- (N = 16)
	114	R	R (19/19)	G (1/75), S (1/75), -- (N = 16)
	115	G	G (19/19)	R (1/75), -- (N = 16)
	116	P	P (19/19)	S (1/75), -- (N = 16)
	117	S	S (19/19)	F (2/75), -- (N = 16)
	118	F	F (19/19)	S (1/76), P (1/76), N (1/76), -- (N = 15)
	119	K	K (19/19)	T (4/78), N (2/78), Q (1/87), -- (N = 13)
	120	P	P (19/19)	K (1/79), A (1/79), L (1/79), -- (N = 12)
	121	Y	Y (19/19)	S (1/83), -- (N = 8)
	122	G	G (19/19)	V (1/83), A (1/83), -- (N = 8)
	123	G	G (19/19)	E (1/84), -- (N = 7)
	124	T	T (19/19)	A (1/84), G (1/84), -- (N = 7)
	125	A	A (19/19)	T (1/90), V (1/90), D (1/90), G (1/90), -- (N = 1)
	126	Y	Y (19/19)	K (1/90), H (1/90), -- (N = 1)
	135	F	F (19/19)	L (1/91)
	140	I	I (19/19)	V (10/91), L (3/91)
	141	D	D (19/19)	G (1/91), Q (1/91), E (1/91)
	142	T	T (19/19)	S (2/91), A (1/91)
	143	G	G (19/19)	E (11/91), D (2/91)
HVR4 (162–167)	162	S	S (19/19)	G (7/91), A (2/91) T (1/91)
	163	A	A (19/19)	N (3/91)
	164	K	K (19/19)	N (2/91), Q (1/91)
	165	D	D (19/19)	T (3/91)
	166	K	K (18/19), T (1/19)	N (3/91), E (1/91), T (1/91)

Abbreviations: --: unavailable; A: alanine; E: glutamic acid, isoleucine; V: valine; R: arginine; Q: glutamine; L: leucine; P: proline; N: asparagine; G: glycine; K: lysine; S: serine; D: aspartic acid; H: histidine; F: phenylalanine; T: threonine; Y: tyrosine. These hypervariable regions (HVR1-4) were determined as previously described [19].

## Data Availability

The original contributions presented in this study are included in the article/Appendix A. Further inquiries can be directed to the corresponding author.

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
