# Peer review of "Widespread Detection of Fowl Adenovirus Serotype 2/11 Species D Among Cases of Inclusion Body Hepatitis–Hydropericardium Syndrome in Chickens in Egypt"

_microorganisms, 2025, doi:10.3390/microorganisms13051107_

Round 1

Reviewer 1 Report

Comments and Suggestions for Authors

Doaa M. Abdellatif et al. identify the circulating FAdVs from cases of IBH-HPS in 5 Egyptian provinces during the period from October 2020 through September 2022,phylogenetic analysis of 19 strains revealed clustering into FAdV species D serotype 2/11 demonstrating that serotype 2/11 is most prevalent in the targeted Egyptian provinces.  The research is of great significance and can provide a very clear understanding of the prevalence of poultry diseases in Egypt.

  1. Previous studies in Egypt reported FAdV serotypes belonging to different species A and B, Now it has transformed into serotypes 2/11.Whether this transformation is due to immune stress or the influence of other countries, the author needs to point it out in the discussion.
  2. IBH-HPS outbreaks were reported in many countries globally, such as Japan, China,

Iran, India, Malaysia, Spain, Saudi Arabia, and South Africa. It is suggested that the author increase the literature support

  1. Compared with the 8a , whether the virulence Egyptian serotype 2/11 isolates has increased or decreased. The author didn't point it out.
  2. FAdVs with seven hypervariable regions (HVRs 1-7) Genetic variation. It is suggested that the author could use molecular simulation to make it more intuitive

Author Response

1. Previous studies in Egypt reported FAdV serotypes belonging to different species A and B, Now it has transformed into serotypes 2/11. Whether this transformation is due to immune stress or the influence of other countries, the author needs to point it out in the discussion. Fowl adenovirus serotypes 1 and 3, belonging to species A and B respectively, were isolated from only two flocks and have been reported just once in a single study conducted in Egypt. In contrast, the majority of studies—including ours—have consistently identified serotype 2/11 of species D, which appears to be more prevalent and widely distributed. To enhance clarity, this section has been rewritten; please refer to paragraph 6 in the Discussion. 2. IBH-HPS outbreaks were reported in many countries globally, such as Japan, China, Iran, India, Malaysia, Spain, Saudi Arabia, and South Africa. It is suggested that the author increase the literature support. Thank you for your valuable comment. We have strengthened the literature support by incorporating references 4–7, as noted in paragraph 3 of the Introduction. 3. Compared with the 8a, whether the virulence of Egyptian serotype 2/11 isolates has increased or decreased. The author didn't point it out. Mutations in the sequenced region of the hexon protein gene were reported in all Egyptian strains available in the NCBI database and tabulated in (Table 4), however, to the best of our knowledge, no experimental studies were conducted to determine the effect of these point mutations on pathogenicity and or immunogenicity of FAdv serotypes circulating in Egypt. This is planned to be our future studies as highlighted in the discussion (paragraph 10) 4. FAdVs with seven hypervariable regions (HVRs 1-7) Genetic variation. It is suggested that the author could use molecular simulation to make it more intuitive. Thank you for your valuable input. We agree that molecular simulation could provide a more intuitive understanding of the genetic variation within the seven hypervariable regions (HVRs 1–7) of FAdVs. While such simulations are beyond the current scope of this study, we are actively seeking funding to explore this further. Nonetheless, the current research lays the groundwork for future studies by identifying key genetic variations and their potential implications for vaccine development and pathogenicity. We believe these findings provide a strong foundation for incorporating molecular modeling and structural analysis in subsequent investigations.

Reviewer 2 Report

Comments and Suggestions for Authors

This paper reports a study aimed at identifying circulating foul adenoviruses in Egypt. The authors were able to identify foul adenoviruses in the 66 flocks out of 210 examined. Hepatitis that featured basophilic inclusion bodies was a common lesion.

A conventional polymerase chain reaction targeting loop 1 of the major hexon gene was used to detect the adenoviruses. Sanger sequencing was used to produce sequences that were used to construct a phylogenetic analysis of 19 strains.

This study was able to highlight the importance of foul adenovirus as a cause of substantial mortality in broiler flocks in Egypt and will provide useful information on the choice of vaccine strains that can be used to control this disease.

Specific comments:

While this study does an excellent job of reporting the presence of these viruses and highlighting the importance of the outbreaks I’m a little concerned at the authors may not fully appreciate the role played by vertical transmission from broiler breeder flocks in initiating these outbreaks.

The references (Hess, 2017) and (Schachner et al., 2018) provide useful background information. However, more recent papers provide additional information at the authors may find useful including (De Luca and Hess, 2025, El-Shall et al., 2022, Franzo et al., 2023). The authors do quote (Franzo et al., 2023) and this paper alludes to the very conflicting epidemiological picture that complicates the diagnosis of fowl adenovirus infections in broilers.

When a naïve flock of broiler breeders is infected there is a cascade of events that follow. Early in the infection limited numbers of birds are infected and there is vertical transmission of the adenoviruses to the day old progeny. The majority of birds on the broiler flock have not received maternal antibody and are very susceptible to the infection. There is also good evidence that infection of broilers less than seven days of age produces a much more severe infection than that seen in older birds.

As the outbreak progresses in the breeder birds the number of birds that transfer maternal antibody as IgY to the progeny chickens increases and this provides substantial protection to some of these birds and slows down the transmission within the flock. This results in much lower mortality and in many cases infection at a slightly older age with reduced mortality.

The eventual recognition of the problem results in diligent application of vaccines to the breeder flocks. However, with time the perceived need for vaccination is reduced and eventually unvaccinated flocks become infected. This results in a new cycle of infection every few years making an understanding of the epidemiology much more difficult.

In the discussion it would be helpful for the authors to give a little more emphasis to the need for vaccination with the appropriate strains of viruses over quite a long period. There is also a need to continue to monitor the changes in genotype prompting further changes in the selection of appropriate vaccines.

In the abstract it was noted that “Positive samples demonstrated pathognomonic histopathological lesions for FAdVs including basophilic and, sometimes, eosinophilic intra-nuclear inclusion bodies”  This statement needs to be rewritten as clearly samples cannot participate in demonstrating lesions. A study was carried out and the lesions were observed.

On page 3 the authors make the statement “Secondly, vaccination failure due to the immunosuppressive nature of FAdVs and the negative effect of FAdVs on feed conversion and weight gain, which in turn increases the carcass condemnation rates at slaughter” I have difficulty in determining how the study by (El-Tholoth and Abou El-Azm, 2019) justifies this statement.

In the next paragraph the authors indicate that infected birds showed lesions. Again these birds are not capable of showing anything. Lesions were observed but the birds did not contribute to demonstrating this situation.

On page 6 it was noted that portal vessels showed severe dilation and congestion. The dilation and congestion was observed but the vessels did not show it.

The caption for Figure 3 should also be rewritten to indicate that these lesions were observed.

Similarly on page 10 it was noted that a sample will be considered positive if it exhibits pathological alterations in the hepatic tissue. This should be rewritten to note that it was considered positive if these lesions are observed.

The second last paragraph on page 11 it is the waning of maternal antibody not the weaning of the antibody.

While Hess in 2013 suggests that maternal antibody may not be fully protective. He was basing this on vaccines available at the time. Other studies have shown that the combination of live vaccination followed by killed vaccines may produce higher levels of maternal antibody in the progeny and this can equate to improved protection of these birds.

On page 11 in the first paragraph the authors probably meant “reported in Egypt” not reported Egypt

In the same paragraph there’s something missing in the quote denoting the widespread of

In the third paragraph it is noted that antigenic determinants present on the Hexon surface are solely in charge of triggering an immune response. The immune response to the adenoviruses can be the result of numerous epitopes throughout the virus that result in both antibody and cell-mediated immune responses. The authors may have been referring to neutralising antibodies. If this is the case, they should be much more specific.

DE LUCA, C. & HESS, M. 2025. Vaccination strategies to protect chickens from fowl adenovirus (FAdV)-induced diseases: A comprehensive review. Vaccine, 43, 126496.

EL-SHALL, N. A., EL-HAMID, H. S. A., ELKADY, M. F., ELLAKANY, H. F., ELBESTAWY, A. R., GADO, A. R., GENEEDY, A. M., HASAN, M. E., JAREMKO, M., SELIM, S., EL-TARABILY, K. A. & EL-HACK, M. E. A. 2022. Epidemiology, pathology, prevention, and control strategies of inclusion body hepatitis and hepatitis-hydropericardium syndrome in poultry: A comprehensive review. Front Vet Sci, 9, 963199.

EL-THOLOTH, M. & ABOU EL-AZM, K. I. 2019. Molecular detection and characterization of fowl adenovirus associated with inclusion body hepatitis from broiler chickens in Egypt. Trop Anim Health Prod, 51, 1065-1071.

FRANZO, G., FAUSTINI, G., TUCCIARONE, C. M., PASOTTO, D., LEGNARDI, M. & CECCHINATO, M. 2023. Conflicting Evidence between Clinical Perception and Molecular Epidemiology: The Case of Fowl Adenovirus D. Animals (Basel), 13.

HESS, M. 2017. Commensal or pathogen - a challenge to fulfil Koch's Postulates. Br Poult Sci, 58, 1-12.

SCHACHNER, A., MATOS, M., GRAFL, B. & HESS, M. 2018. Fowl adenovirus-induced diseases and strategies for their control - a review on the current global situation. Avian Pathol, 47, 111-126.

Comments on the Quality of English Language

Tissues and birds do not show lesions. These lesions are observed.

Author Response

1. While this study does an excellent job of reporting the presence of these viruses and highlighting the importance of the outbreaks I’m a little concerned that the authors may not fully appreciate the role played by vertical transmission from broiler breeder flocks in initiating these outbreaks. The study already reported that 3 one-day-old broiler chicken flocks were found positive by PCR, which indicates a potential role of the vertical transmission of FAdVs from broiler breeders. We further discuss (Discussion paragraph 5). 2. The references (Hess, 2017) and (Schachner et al., 2018) provide useful background information. However, more recent papers provide additional information at the authors may find useful including (De Luca and Hess, 2025, El-Shall et al., 2022, Franzo et al., 2023). The authors do quote (Franzo et al., 2023) and this paper alludes to the very conflicting epidemiological picture that complicates the diagnosis of fowl adenovirus infections in broilers. 3. Thank you for your comment. We included the reference to Hess (2017) because it discusses the primary pathogenicity of FAdVs. In response to your suggestion, we have updated the introduction section and incorporated more recent references, which have enriched both the Introduction and Discussion sections. Additionally, we have replaced the reference to Schachner et al. (2018) with De Luca and Hess (2025) in paragraph 4 of the Introduction to reflect more current findings. 4. When a naïve flock of broiler breeders is infected there is a cascade of events that follow. Early in the infection limited numbers of birds are infected and there is vertical transmission of the adenoviruses to the day old progeny. The majority of birds on the broiler flock have not received maternal antibody and are very susceptible to the infection. There is also good evidence that infection of broilers less than seven days of age produces a much more severe infection than that seen in older birds. As the outbreak progresses in the breeder birds the number of birds that transfer maternal antibody as IgY to the progeny chickens increases and this provides substantial protection to some of these birds and slows down the transmission within the flock. This results in much lower mortality and in many cases infection at a slightly older age with reduced mortality. The eventual recognition of the problem results in the diligent application of vaccines to the breeder flocks. However, with time the perceived need for vaccination is reduced and eventually unvaccinated flocks become infected. This results in a new cycle of infection every few years making an understanding of the epidemiology much more difficult. In the discussion, it would be helpful for the authors to give a little more emphasis to the need for vaccination with the appropriate strains of viruses over quite a long period. There is also a need to continue to monitor the changes in genotype prompting further changes in the selection of appropriate vaccines. We appreciate the reviewer’s insightful observation and have expanded our discussion to address the relevant points. Specifically, we have linked our findings to the following aspects: • The role of maternal antibodies – addressed in Discussion, paragraph 5 • The role of vaccination in preventing FAdV circulation – addressed in Discussion, paragraph 9 • The importance of continued monitoring of genotype changes – addressed in Discussion, paragraph 9 5. In the abstract, it was noted that “Positive samples demonstrated pathognomonic histopathological lesions for FAdVs including basophilic and, sometimes, eosinophilic intra-nuclear inclusion bodies” This statement needs to be rewritten as clearly samples cannot participate in demonstrating lesions. A study was carried out and the lesions were observed. As per the reviewer’s suggestion, we have revised the statement. 6. On page 3 the authors make the statement “Secondly, vaccination failure due to the immunosuppressive nature of FAdVs and the negative effect of FAdVs on feed conversion and weight gain, which in turn increases the carcass condemnation rates at slaughter” I have difficulty in determining how the study by (El-Tholoth and Abou El-Azm, 2019) justifies this statement. Thank you for your thorough review. We acknowledge that the referenced study by El-Tholoth and Abou El-Azm (2019) does not directly support the statement in question, which was related to FAdV-induced adenoviral gizzard erosion (AGE). To ensure clarity and accuracy, we have removed the statement from the manuscript. 7. In the next paragraph the authors indicate that infected birds showed lesions. Again these birds are not capable of showing anything. Lesions were observed but the birds did not contribute to demonstrating this situation. We have revised the statement and made the necessary changes throughout the manuscript. 8. On page 6 it was noted that portal vessels showed severe dilation and congestion. The dilation and congestion was observed but the vessels did not show it. While we acknowledge that suboptimal eosin staining may have affected the visibility of red coloration indicating congestion in the image, we have removed the word “severe” from the figure captions for Figures 2 and 3 to clarify this limitation. 9. Similarly on page 10 it was noted that a sample will be considered positive if it exhibits pathological alterations in the hepatic tissue. This should be rewritten to note that it was considered positive if these lesions are observed. The statement has been revised to clearly indicate that these lesions were observed, as reflected in Discussion, paragraph 3. 10. The second last paragraph on page 11 it is the waning of maternal antibody not the weaning of the antibody. Thank you for your careful review. The word “weaning” was replaced by “waning” in the revised manuscript. 11. While Hess in 2013 suggests that maternal antibodies may not be fully protective. He was basing this on vaccines available at the time. Other studies have shown that the combination of live vaccination followed by killed vaccines may produce higher levels of maternal antibody in the progeny and this can equate to improved protection of these birds. We appreciate your insightful suggestion, We have revised the Discussion accordingly to incorporate this perspective, please refer to Discussion, paragraph 9. 12. On page 11 in the first paragraph, the authors probably meant “reported in Egypt” not reported Egypt. The typo was corrected. 13. In the same paragraph there’s something missing in the quote denoting the widespread of we meant that the widespread circulation of FAdV serotype 2/11in Egypt. The typo was corrected. 14. In the third paragraph it is noted that antigenic determinants present on the Hexon surface are solely in charge of triggering an immune response. The immune response to the adenoviruses can be the result of numerous epitopes throughout the virus that result in both antibody and cell-mediated immune responses. The authors may have been referring to neutralising antibodies. If this is the case, they should be much more specific. Thank you for this important clarification. The statement has been revised to enhance clarity and specify that the antigenic determinants on the Hexon surface are primarily associated with the induction of neutralizing antibodies. Please refer to paragraph 8 in the Discussion for the updated wording. 15. DE LUCA, C. & HESS, M. 2025. Vaccination strategies to protect chickens from fowl adenovirus (FAdV)-induced diseases: A comprehensive review. Vaccine, 43, 126496. EL-SHALL, N. A., EL-HAMID, H. S. A., ELKADY, M. F., ELLAKANY, H. F., ELBESTAWY, A. R., GADO, A. R., GENEEDY, A. M., HASAN, M. E., JAREMKO, M., SELIM, S., EL-TARABILY, K. A. & EL-HACK, M. E. A. 2022. Epidemiology, pathology, prevention, and control strategies of inclusion body hepatitis and hepatitis-hydropericardium syndrome in poultry: A comprehensive review. Front Vet Sci, 9, 963199. EL-THOLOTH, M. & ABOU EL-AZM, K. I. 2019. Molecular detection and characterization of fowl adenovirus associated with inclusion body hepatitis from broiler chickens in Egypt. Trop Anim Health Prod, 51, 1065-1071. FRANZO, G., FAUSTINI, G., TUCCIARONE, C. M., PASOTTO, D., LEGNARDI, M. & CECCHINATO, M. 2023. Conflicting Evidence between Clinical Perception and Molecular Epidemiology: The Case of Fowl Adenovirus D. Animals (Basel), 13. HESS, M. 2017. Commensal or pathogen - a challenge to fulfil Koch's Postulates. Br Poult Sci, 58, 1-12. SCHACHNER, A., MATOS, M., GRAFL, B. & HESS, M. 2018. Fowl adenovirus-induced diseases and strategies for their control - a review on the current global situation. Avian Pathol, 47, 111-126. As per the reviewer's suggestions, we included the above references in the manuscript